# X-Prompt: Multi-modal Visual Prompt for Video Object Segmentation

## ABSTRACT

Multi-modal Video Object Segmentation (VOS), including RGB-Thermal, RGB-Depth, and RGB-Event, has garnered attention due to its capability to address challenging scenarios where traditional VOS methods struggle, such as extreme illumination, rapid motion, and background distraction. Existing approaches often involve designing specific additional branches and performing full-parameter fine-tuning for fusion in each task. However, this paradigm not only duplicates research efforts and hardware costs but also risks model collapse with the limited multi-modal annotated data. In this paper, we propose a universal framework named X-Prompt for all multi-modal video object segmentation tasks, designated as RGB+X. The X-Prompt framework first pre-trains a video object segmentation foundation model using RGB data, and then utilize the additional modality of the prompt to adapt it to downstream multi-modal tasks with limited data. Within the X-Prompt framework, we introduce the Multi-modal Visual Prompter (MVP), which allows prompting foundation model with the various modalities to segment objects precisely. We further propose the Multi-modal Adaptation Expert (MAEs) to adapt the foundation model with pluggable modality-specific knowledge without compromising the generalization capacity. To evaluate the effectiveness of the X-Prompt framework, we conduct extensive experiments on 3 tasks across 4 benchmarks. The proposed universal X-Prompt framework consistently outperforms the full fine-tuning paradigm and achieves state-of-the-art performance. Codes will be available.

## CCS CONCEPTS

• **Computing methodologies → Video segmentation**.

## KEYWORDS

Multi-modal Video Object Segmentation, X-Prompt, Multi-modal Visual Prompt, Multi-modal Adaptatin Expert, RGB-X

## 1 INTRODUCTION

Video Object Segmentation (VOS) [4, 5, 21, 22, 54, 57, 76, 90] is a pivotal task designed to delineate target objects throughout a video sequence. This paper primarily delves into semi-supervised VOS, a foundational endeavor that aims to segment a specified object across an entire video sequence, leveraging the object mask provided in the initial frame. This task is instrumental for a myriad of applications,

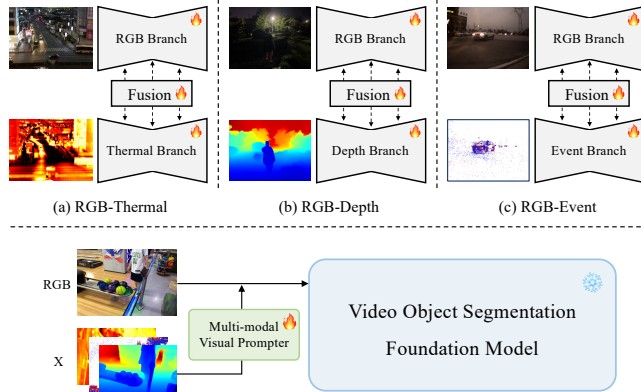

**Figure 1: (a), (b), and (c) current RGB-T, RGB-D, and RGB-E video object segmentation paradigms. (d) X-Prompt, a unified framework for RGB+X multi-modal video object segmentation tasks.**

such as video editing [42, 50, 67], surveillance [1, 43, 52, 89], robotics [20], and autonomous driving [33, 83, 86]. Recent studies, particularly those employing matching-based methods [7, 29, 63, 78, 81, 92] even enhanced by memory mechanisms [16, 23, 25, 51, 58–60], have made significant progress and achieved advanced results. These methods assess similarities between previous and current frames through semantic matching at the pixel level. However, despite these efforts and advancements, contemporary state-of-the-art methodologies still struggle in challenging scenarios like extreme illumination, obscured scenes, rapid motion, occlusions, and background distraction.

Alternative modalities offer a vital solution to these limitations, by serving as a crucial complement to RGB data in challenging scenarios. For example, thermal [35, 62] and depth [46] information remains unaffected by variations in illumination and visibility conditions. Similarly, event data [18], which records changes in pixel intensity at high frequencies, does not blur during rapid motion. Consequently, Multi-modal Video Object Segmentation is garnering increased interest for its capacity to facilitate more resilient tracking through the exploitation of inter-modal complementarities, including combinations like RGB-Thermal (RGB-T) [73, 85], RGB-Depth (RGB-D) [74, 88], and RGB-Event (RGB-E) [36, 66]. Current multi-modal VOS approaches predominantly augment an existing VOS model's RGB branch with an additional modality branch and a fusion module. These enhancements are then subjected to either full training or fine-tuning on downstream tasks, as depicted in Fig. 1 (a), (b), and (c). However, on one hand, while all these modifications aim to improve VOS task performance in complex scenarios

 

by introducing additional modal information, the branches and fusion mechanisms are tailored separately for each task, necessitating different structures for each downstream modality. This paradigm not only duplicates research efforts but also escalates the computational and hardware costs associated with developing specialized architectures for every task. On the other hand, full-parameters training or fine-tuning these newly designed architectures with limited data can easily lead to overfitting, alongside the risk of catastrophic forgetting, thus lose the generalized capabilities of the original pretrained model. Moreover, the collection of multi-modal paired data and pixel-level annotations remains challenging and costly, resulting in a scarcity of such data, with each modality having approximately only a hundred or so annotated sequences. Training with such limited data restricts the scale of the model, making it difficult to effectively train the new modality branch and the fusion module without degrading or collapsing the performance and the generalizability of the pretrained model, thereby failing to leverage prior knowledge.

To address these challenges, we argue that these multi-modal tasks of RGB-T, RGB-D, and RGB-E can be unified into a single effort, **RGB-X**, where X represents any additional modality. And they can share the common segmentation capability and benefit from the robust generalization of a video object segmentation foundation model, which can segment target objects in the current frame based on reference frames and their segmentation masks in various videos. Therefore, we propose the **X-Prompt** framework, a universal solution for multi-modal video object segmentation. This framework utilizes numerous RGB video sequences to pre-train a transformer-based foundation model with general object segmentation capacity. Subsequently, additional modality information is utilized to prompt and adapt this foundation model to downstream multi-modal tasks with limited data while retaining the model's generalization ability.

Within the X-Prompt framework, we further propose two key components: the Multi-modal Visual Prompter and the Multi-modal Adaptation Experts. The **Multi-modal Visual Prompter (MVP)** encodes cross-modality information into visual prompts that used to prompt the frozen foundation model. The spatial-channel attended complementary prompt are integrated with the patch embeddings of the foundation model to guide segmentation of downstream modality. Additionally, multi-scale multi-modal prompts are obtained to prompt the segmentation mask decoder to incorporate diverse cross-modality information, enhancing the precision of object delineation in this dense prediction tasks. The design described above enables the foundation model to leverage cross-modality prompts to achieve RGB-X VOS tasks, but it does not allow the foundation model to learn new modality-specific knowledge without any tuning. However, full-parameter fine-tuning poses a risk of degrading or collapsing the foundation model, especially with limited data. Therefore, we introduce the **Multi-modal Adaptation Experts (MAEs)** for plugging modality-specific knowledge without forgetting the generalized prior knowledge embedded in the frozen foundation model. Within each transformer layer of the frozen foundation model, we employ low-rank adaptations to serve as the modality experts. These experts' outputs are combined via a router to facilitate multi-modal collaboration. This approach ensures that the prior knowledge of the foundation model is retained

while integrating additional expertise required for new tasks, such as extracting new modality patterns including feature extraction and matching, as well as facilitating collaborative usage of RGB with other modalities. The proposed universal X-Prompt Framework achieves state-of-the-art performance on 3 multi-modal video object segmentation tasks including RGB-T, RGB-D, and RGB-E, surpassing existing methods significantly. In summary, our contributions are as follows:

- We propose X-Prompt, the first universal framework for multi-modal video object segmentation, including RGB-T, RGB-D, and RGB-E tasks. It utilizes the X-modality as the visual prompt to adapt a foundation model to various downstream tasks.
- We designed the Multi-modal Visual Prompter, which encodes visual prompts into both the foundation model and the mask decoder, enabling precise objects segmentation across various modalities.
- We present the Multi-modal Adaptation Expert to adapt the foundation model with pluggable modality-specific knowledge without forgetting generalized prior knowledge.
- The proposed X-Prompt framework achieves state-of-the-art performance on 3 multi-modal video object segmentation tasks, including RGB-T, RGB-D, and RGB-E.

## 2 RELATED WORK

### 2.1 Video Object Segmentation

Semi-supervised Video Object Segmentation (VOS) accurately segments objects in a video sequence using the provided mask in initial frame. Model-based methods [2, 3, 48, 69] start with offline pre-training on static images to capture segmentation features, then refining them using mask from the first frame of the test video. However, the time-consuming fine-tuning step limits their suitability for real-world applications. Then, Propagation-based methods [10, 28, 31, 50, 53, 65] iteratively propagate masks with temporal correlations. However, these methods heavily rely on previous segmentation masks, making them susceptible to target drift and error accumulation. To address these drawbacks, Matching-based methods [7, 29, 63, 78, 81, 92] calculate correspondence pixel maps between the current and reference frames. Among them, FEELVOS [63] CFBI [78] and CFBI+ [81] improve it by integrating foreground-background features and employing local-global matching. More recently, memory-based methods [16, 23, 25, 51, 58–60] propose an external memory bank to store history features, thus addressing contextual limitations. Techniques such as XMem [8] extend memory to long-term VOS and AOT [77], DeAOT [82] enhancing performance through Transformer. Despite advancements in existing methods, challenges persist particularly in scenarios with background clutter, illumination variation, and motion blur. This paper tackles these challenges by emphasizing multi-modal VOS, utilizing diverse data sources beyond RGB inputs.

### 2.2 Multi-modal Tracking and Segmentation

In video object tracking, researchers have explored using multiple modalities to improve accuracy. For instance, [44] and [49] use depth maps for handling occlusions. DAL [56] introduces embedded depth-aware convolution into RGB tracking frameworks to

bolster target estimation robustness. Besides, [40] combines RGB and event flows to predict Regions of Interest (ROIs) in complex scenarios. To leverage dual modalities, MDNet [66] merge visible frames and event flows onto RGB tracking frameworks to construct a series of baseline trackers. Similarly, JMMAC [84] integrates RGB and thermal infrared modalities to capture reliable appearance and motion cues, with event flows significantly contributing to robust tracking. More recently, to tackle severe data deficiencies, some studies have proposed data-efficient methods. For instance, Pro-Track [75] transfers multi-modal inputs to a single modality, while ViPT [91] designs modality-complementary prompts. As a more challenging task, multi-modal VOS becomes increasingly important as it provides precise pixel-level masks, critical for applications like surveillance in adverse weather and nighttime autonomous driving. VTiNet [73] tackles Visible-Thermal VOS by integrating cross-modal and cross-frame features in modality and temporal domains, alongside a modality-sensitive memory bank. [36] improves low-light VOS with event assistance through the adaptive cross-modal fusion and event-guided memory matching modules. Notely, FusedCDNet [74] proposes a weakly-supervised RGB-D VOS method, utilizing bounding box level supervision in both training and testing phases, mitigating challenges of expensive data collection and annotation. To address the issue of model generalization degradation due to limited data, as well as the redundant research effort and computational deployment costs resulting from designs tailored for each specific modal task, this paper proposes a universal framework for all multi-modal VOS tasks. It utilizes the auxiliary modality as the prompt to adapt a foundational model.

## 2.3 Foundation Model Adaptation

The advancement of foundation models has prompted exploration into their adaptation for downstream tasks. Recently, freezing the pre-trained model and only fine-tune a few additional parameters to attain strong performance has emerged as a notable approach, which is particularly valuable when data of downstream task is limited. Among them, prompt tuning-based methods [34] fine-tune models by introducing learnable prompt tokens. For instance, VPT [30] enhances transformer encoders with learnable parameters, outperforming full fine-tuning on 20 downstream recognition tasks. AdaptFormer [6] investigates efficient fine-tuning in video action recognition by integrating lightweight modules into a ViT, surpassing the performance of existing fully fine-tuned models. ProTrack [75] and ViPT [91] utilize prompt tuning in multimodal tracking to address challenging scenarios. Additionally, adapter-based methods [26] incorporate trainable adapters into pre-trained models. For example, LoRA [27] injects trainable low-rank matrices into transformer layers to approximate the weight updates and $(IA)^3$ [41] scales activations by learned vectors to attain stronger performance with a relatively tiny amount of new parameters. This paper not only uses multi-modal visual prompts to enable the foundation model to handle various multi-modal tasks, but also employs modality adaptation experts to introduce knowledge about new modalities to the foundation model.

## 3 METHOD

In this section, we begin by formulating of multi-modal video object segmentation tasks under a unified formulation termed RGB-X

(Sec. 3.1.1). We then introduce the video object segmentation foundational model with generalization capabilities (Sec 3.1.2). Following this, we propose the universal X-Prompt framework, which then leverages the additional modality as the prompt to adapt the foundation model to various multi-modal tasks (Sec. 3.2). Within the X-Prompt framework, we further introduce the Multi-modal Visual Prompter that enables prompting the foundation model to segment objects precisely with various modalities (Sec. 3.3) and the Multi-modal Adaptation Experts to adapt the foundational model with pluggable modality-specific knowledge without forgetting generalized prior knowledge (Sec. 3.4).

## 3.1 Preliminary

### 3.1.1 Problem Formulation.

Multi-modal video object segmentation tasks, such as RGB-Thermal, RGB-Depth, and RGB-Event, can be abstracted into a unified task, termed **RGB-X**, which involves leveraging an auxiliary modality in addition to RGB to segment target objects in each frame. To formulate, given a video with $T$ frames denoted as $\{I_t, X_t\}_{t=1}^{T}$, along with the object masks in the first frame $M_1 \in \mathbb{O}^{H \times W}$ that identify $O$ targets of interest, the goal is to predict the object masks in all frames, represented as $\{M_t \in \mathbb{O}^{H \times W}\}_{t=1}^{T}$.

### 3.1.2 Foundation Model.

The models required for multi-modal video object segmentation tasks all hinge on a core capability: computing the correlation between the current frame and the first frame with the known object mask. Therefore, we argue that *RGB-X tasks can share the segmentation capability and the generalization of a foundational model for RGB video object segmentation.* This model should be capable of segmenting target objects in the current frames by referencing frames and their segmentation masks when dealing with diverse videos.

Specifically, we built our foundation model based on the Vision Transformer [15, 19] within an All-in-One Transformer architecture following [37]. To begin, we generate the non-overlapping patch embeddings with Patch Embedding layer for the reference and current frame $\mathbf{P}(I_r)$ and $\mathbf{P}(I_t)$, and concatenate them along the spatial dimension as the initial input of transformer:

$$z^0 = \text{Cat}[\mathbf{P}(I_t), \mathbf{P}(I_r)] \in \mathbb{R}^{(N_t + N_r) \times D}, \quad (1)$$

where $\mathbf{P}$ is the Patch Embedding layer. Supposing $p$ is the stride of the patch embedding, thus $N = HW/p^2$. Then, this input along with the mask of reference frame are fed into Foundation Model:

$$z_t^L = \mathbf{F}\left(z_t^0, z_r^0, M_r\right) \quad (2)$$

where $\mathbf{F}$ is a Visual Transformer with $L$ layers And in each layer $\mathbf{F}^{(l)}$, $l \in \{1, 2, \cdots, L\}$, the forward process of each layer $z^l = \mathbf{F}^{(l)}\left(z^{l-1}, m_r\right)$ can be written as:

$$z'^{l-1} = \text{LN}\left(\text{MSA}\left(z^{l-1}, m_r\right)\right) + z^{l-1},$$
$$z^l = \text{LN}\left(\text{FFN}\left(z'^{l-1}\right)\right) + z'^{l-1}, \quad (3)$$

where LN denotes the Layer Norm and FFN represents the Feed-Forward Network. MSA can use the naive multi-head self attention

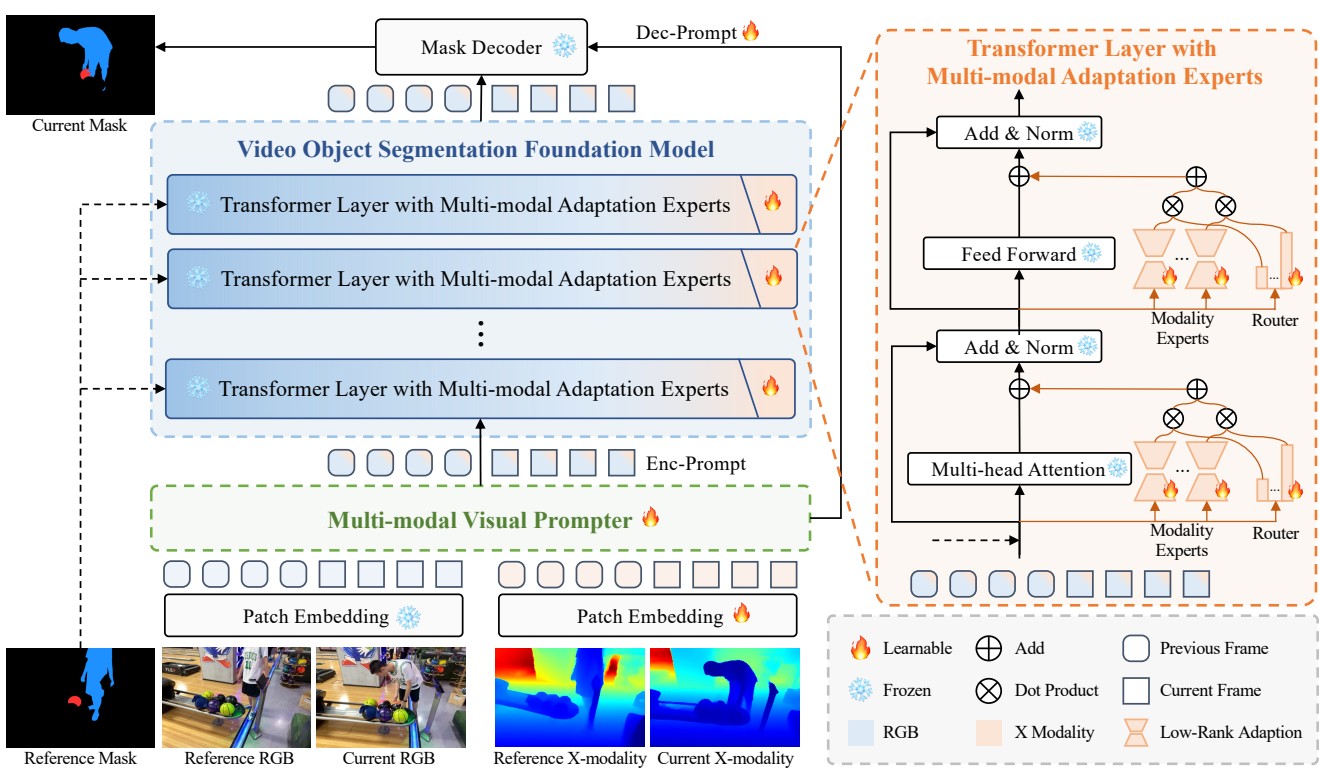

**Figure 2: The overall architecture of universal X-Prompt Framework for RGB-X multi-modal video object segmentation tasks. Following the pre-training of an RGB VOS foundation model (Sec. 3.1) with robust segmentation capabilities and generalization, X-Prompt (Sec. 3.2) utilizes the X-modality to prompt and adapt the foundation model for various downstream multi-modal tasks, employing our proposed Multi-modal Visual Prompter (Sec. 3.3) and Multi-modal Adaptation Experts (Sec. 3.4).**

or the specifically designed uni-hybrid attention in [37]. For integration with mask information, the query, key, and value are:

$$\{Q_r^l, \ K_r^l, \ V_r^l\} = \{W_q^l z_r^l, \ W_k^l z_r^l, \ W_v^l z_r^l + m_r\}, \quad (4)$$

where $W_q^l$, $W_k^l$, and $W_v^l$ are projection weights for query, key, and value transformations. $m_r \in \mathbb{R}^{N_r \times D}$ is the mask-embedding generated through a mask embedding layer through a single convolutional layer. Finally, the output $z^L$ of the foundation model, is input to the lightweight mask decoder $\mathbf{D}$ and decoded into final multi-object masks $M_t = \mathbf{D}\left(z^L\right)$. We have included only one reference frame here for brevity. Actually, multiple reference frames are dynamically updated and stored in memory. We refer readers to [37] for more design details about our video object segmentation foundation model.

We train the foundation model with both synthetic video sequences generated from static RGB image datasets [11, 17, 24, 39, 61] and various RGB VOS datasets [14, 55, 70]. This pretraining on large-scale datasets equips our video object segmentation foundation model with robust temporal matching capabilities and facilitates effective transferability to downstream multi-modal RGB+X VOS tasks.

## 3.2 X-Prompt Framework

Multi-modal video object segmentation tasks involve an additional auxiliary X-modality flow that is temporally synchronized and spatially aligned with the RGB flow. We propose the X-Prompt framework, which first trains a robust RGB video object segmentation foundation model with strong segmentation capabilities and generalization. Subsequently, we utilize the auxiliary modality flow as the prompt to adapt the model to downstream multi-modal tasks with limited data. As illustrated in Fig. 2, in the X-Prompt framework, the RGB images $I$ and the X-modality maps $X$ are feed into their respective patch embedding layers $\mathbf{P}_i$ and $\mathbf{P}_x$ to obtain patch tokens:

$$z_{\text{rgb}}^{16\times} = \mathbf{P}_i\left(I\right), z_x^{16\times} = \mathbf{P}_x\left(X\right), \in \mathbb{R}^{(N_t+N_r)\times D}, \quad (5)$$

where $N_t$ and $N_r$ are the number of patch tokens of the current frame and the reference frames. Then the X-modality tokens are utilized to supplement the image tokens, resulting in complementary combined multi-modal prompts embedding through the Multi-modal Visual Prompter (detailed in Sec. 3.3):

$$z^0 = \Phi\left(z_{\text{rgb}}^{16\times}, z_x^{16\times}\right) \in \mathbb{R}^{(N_t+N_r)\times D}, \quad (6)$$

where $z^0$ is the spatial-channel attended multi-modal prompts and $\Phi$ indicates the proposed Multi-modal Visual Prompter. This visual

**Figure 3: The design of the Multi-modal Visual Prompter (MVP) for encoding the spatial-channel attended complementary prompt embedding for the foundation model and the multi-scale multi-modal prompt embedding for the mask decoder.**

prompt to the foundation model operates with a stride of 16. Additionally, for the pixel-level dense prediction task, we also design the multi-scale multi-modal prompts to mask decoder to accurately segment object masks. Subsequently, the multi-modal prompt $z^0$ is fed to the foundation model, enabling multi-modal video object segmentation without requiring any modifications to the RGB foundation model:

$$z_t^L = \mathbf{F}\left(z^0, M_r\right) \in \mathbb{R}^{N_t \times D}, \tag{7}$$

where $z_t^L$ then can be decoded to the object segmentation mask $M_t$ with the mask decoder $\mathbf{D}$.

Moreover, to adapt the foundation model with modality-specific knowledge in a pluggable manner without compromising its generalization capacity, we introduce Multi-modal Adaptation Experts (detailed in Sec. 3.4) to the frozen foundation model:

$$\hat{z}^l = \Psi\left(\mathbf{F}^{(l)}\left(z^{l-1}, m_r\right)\right), \tag{8}$$

where $\hat{z}^l$ is the output of the $l$-th block adapted with modality experts and $\Psi$ represents the proposed adaptation experts in Sec. 3.3. It's worth noting that within the X-Prompt framework, only the newly introduced prompter, modality experts, and the patch embedding layer for the X-modality are trainable. All other network parameters, including the transformer encoder layers in the foundation model, the mask decoder, and the patch embedding layer for the RGB image, remain frozen from pretraining.

### 3.3 Multi-modal Visual Prompter

To leverage the powerful segmentation capabilities and generalization of foundation models for multi-modal video object tasks, we propose the Multi-modal Visual Prompter (MVP) to encode RGB-X information into visual prompts. To accommodate dense prediction tasks, we employ convolutional patch embedding layers with progressive downsampling to embed patch tokens. This progress yields multi-scale RGB image patch tokens $z_{rgb}^{4\times}$, $z_{rgb}^{8\times}$, and $z_{rgb}^{16\times}$, as well as X-modality patch tokens $z_x^{4\times}$, $z_x^{8\times}$, and $z_x^{16\times}$. We use $z_{rgb}^{16\times16}$ and $z_x^{16\times}$ to encode primary prompts that enable the foundation model to integrate information from both modalities for object tracking and segmentation. The other patch tokens are efficiently encoded

into multi-scale multi-modal prompts to guide the mask decoder for more precise object segmentation masks.

For the visual prompt for the foundation model, given $z_{rgb}^{16\times}$ and $z_x^{16\times}$, we first concatenate them into a single embedding along the specified dimension, followed by a Linear layer that reduces the dimensionality from $2D$ to $D$ to obtain $z_{fuse}^{16\times}$. Subsequently, we obtain spatial attention $A_s \in \mathbb{R}^{\frac{H}{16} \times \frac{W}{16} \times 1}$ and channel attention $A_c \in \mathbb{R}^{1 \times 1 \times D}$ separately, then merge them into spatial-channel attention through broadcast multiplication:

$$A_{sc} = A_s \times A_c \in \mathbb{R}^{\frac{H}{16} \times \frac{W}{16} \times D}, \tag{9}$$

$$A_s = \sigma\left(\text{Conv}\left(\text{AvgPool}\left(z_{fuse}^{16\times}\right)\right) + \text{Conv}\left(\text{MaxPool}\left(z_{fuse}^{16\times}\right)\right)\right), \tag{10}$$

$$A_c = \sigma\left(\text{MLP}\left(\text{AvgPool}\left(z_{fuse}^{16\times}\right)\right) + \text{MLP}\left(\text{MaxPool}\left(z_{fuse}^{16\times}\right)\right)\right), \tag{11}$$

where $\sigma$ denotes the sigmoid function. Conv and MLP represents the convolution layer and the multi-layer perceptron, respectively. Finally, this spatial-channel attention weight is applied to the image path token embedding to produce a multi-modal prompt that is spatial-channel attended, enhancing and complementing the original tokens: $z^0 = A_{sc} \times z_{fuse}^{16\times}$.

For the visual prompt for the mask decoder, we employ a simple, efficient, yet effective Linear Adapter to encode $z_{rgb}^{4\times}$, $z_x^{4\times}$, and $z_{rgb}^{8\times}$, $z_x^{8\times}$ into multi-scale multi-modal prompts $\{z^{4\times}, z^{8\times}\}$ for the mask decoder through residual connection. To be more efficient, we find that the linear layer preceding the upsampling operation in each block of the mask decoder can play the role of the adapter, thus eliminating the need for additional parameters for this step.

### 3.4 Multi-modal Adaptation Experts

Although the proposed MVP effectively encodes multi-modal prompt embeddings for the frozen foundation model, enabling it to handle various multi-modal tasks without degrading the foundation model's performance, the foundation model itself does not learn any new knowledge about X-modality in this process. As a result, the model's performance on each downstream task is sub-optimal. To infuse new knowledge of this modality into the foundation model

using limited data, while avoiding the risks of degrading or collapsing the foundation model that full-parameter fine-tuning might entail, we introduce the Multi-modal Adaptation Experts (MAEs). This approach offers a pluggable knowledge adaptation mechanism that works with the frozen foundation model.

Specifically, within each transformer layer of the frozen foundation model, we employ $K$ low-rank adaptations to act as modality adaptation experts, assisting linear layers of both MSA and FFN in parallel. Here we denote the input tokens to the linear layer of MSA or FFN within the transformer layer, which also are inputs to the experts, as $h \in \mathbb{R}^{(N_t + N_r) \times D}$, to describe how these modality adaptation experts function:

$$
\begin{aligned}
h' &= W_0 \cdot h + \text{Route}_{i=1}^{K} (\triangle h_i) \\
&= W_0 \cdot h + \text{Route}_{i=1}^{K} (\triangle W_i \cdot h)
\end{aligned}
\tag{12}
$$

where $\triangle h_i \in \mathbb{R}^{(N_t + N_r) \times D}$ is the adaptation of each expert. $W_0 \in \mathbb{R}^{D \times D}$ denotes the original parameters of the foundation model. $\triangle W_i \in \mathbb{R}^{D \times D}$ represents the learnable parameters for modality adaptation, implemented efficiently by $\triangle W_i = B_i A_i$ according to the LoRA [27] technique for LLMs. These experts' outputs are combined via a router to facilitate multi-modal collaboration:

$$
\text{Route}_{i=1}^{K} (\triangle h_i) = \text{Softmax} \left( w_i^R \cdot h \right) \sum_{i=1}^{K} \triangle h_i
\tag{13}
$$

where $w_i^R \in \mathbb{R}^{D \times K}$ is the learnable parameter of the Router and is implemented through a linear layer.

This pluggable knowledge adaptation mechanism ensures that general capabilities of the foundation model are retained while integrating additional expertise required for new tasks, such as extracting new modality patterns including feature extraction and matching, as well as facilitating collaborative usage of RGB with X-modality.

# 4 EXPERIMENT

## 4.1 Implementation Details

**Networks.** In our foundation model, following the OneVOS [37], we utilize an All-in-One Transformer architecture. The entire network operates within a ConvMAE [19] structure, where patch embedding layers produce patch tokens at 1/4, 1/8, and 1/16 resolutions through progressively downsampled convolutional layers, effectively preserving multi-scale information. The transformer comprises ten transformer encoder layers that concurrently perform feature extraction and object matching, with each dimension set to 768. The mask decoder is structured as an FPN [38], progressively increasing feature resolution while decreasing the channel dimension. The prompter's convolutional layers utilize a $7 \times 7$ kernel size, and the MLP layer includes one hidden layer with a dimension that is one-sixteenth of the input dimension. The modality adaptation experts are configured with a low-rank of 8, and both the low-rank adaptation and routing are implemented with linear layers. According to our experiments, we typically employ two experts for each multi-modal task.

**Training.** The Foundation model is initialized with ImageNet1k [13] pretrained weights, and is trained on synthetic video sequences generated from still image datasets [11, 17, 24, 39, 61] for 200,000

iterations with a batch size of 4 and a learning rate of 2e-4. Subsequently, it undergoes main training on collected standard VOS datasets including DAVIS [55], YouTube [70], and MOSE [14] for another 200,000 iterations with a learning rate of 1e-4 on four 3090 GPUs. After pre-training, the Foundation model is adapted to multi-modal downstream tasks with all pre-trained parameters frozen, while only the newly introduced prompter, adaptation experts, and the patch embedding layer for the X-modality are trainable. Depending on the number of annotated videos available for each task, the training durations vary from 20,000 to 60,000 iterations. Throughout all stages of training, a consistent 0.5:0.5 combination of bootstrapped cross-entropy loss and soft Jaccard loss is used.

## 4.2 Main Results

X-Prompt accomplishes the consolidation of multiple downstream multi-modal video object segmentation tasks. In this study, we select 4 benchmarks [66, 73, 85, 88] of 3 challenging tasks to assess the effectiveness and generalization of the proposed framework. We conduct modality prompt and adaptation on these downstream tasks without specific modulation. Evaluation metrics include $\mathcal{J}$, $\mathcal{F}$, and $\mathcal{J}\&\mathcal{F}$ [55], where $\mathcal{J}$ represents the Intersection over Union score between the prediction and the ground truth mask, $\mathcal{F}$ denotes the boundary similarity measure between the prediction boundary and the ground truth, and $\mathcal{J}\&\mathcal{F}$ represents the average score of both $\mathcal{J}$ and $\mathcal{F}$.

**RGB-Thermal Task.** To evaluate the performance of the proposed framework on the RGB-Thermal task, we selected the VisT300 [73] and VT-UAV [85] benchmarks. VisT300, a general dataset for RGB-T VOS, includes a variety of scenes, with 250 training sequences and 50 testing sequences. The VT-UAV dataset, designed for the challenging Unmanned Aerial Vehicle (UAV) scenarios, with 50 sequences for training and another 50 for testing. We adapt the training data from these two datasets for the thermal modality task. As Tab. 1 indicates, our proposed X-Prompt outperforms all previous SOTA methods, achieving the highest scores of 84.2% and 87.3% $\mathcal{J}$ and $\mathcal{F}$ on these VisT300 and VT-UAV respectively, especially on the challenging VT-UAV dataset, where it surpasses others by 10.6%.

**RGB-Depth Task.** For the RGB-D task, we use the ARKitTrack to evaluate the performance. ARKitTrack [88] utilizes LiDAR to collect and annotate 300 RGB-D sequences in both static and dynamic scenes, comprising 245 training videos and 55 testing videos. As shown in Tab. 2, after pre-training the foundation model with RGB data and adapting it for the Depth modality using ARKitTrack, our X-Prompt achieves an impressive $\mathcal{J}\&\mathcal{F}$ score of 82.1%, surpassing previous models whether trained exclusively on RGB-D datasets or pre-trained on RGB data.

**RGB-Event Task.** For the RGB-E task, we use VisEvent [66] dataset, the largest visible-event benchmark dataset collected from real-world scenarios for evaluation. However, the VisEvent dataset only provides object annotations at the box level. Therefore, we employed the powerful HQ-SAM [32] method to generate mask annotations based on these box annotations. Through HQ-SAM predictions and our filtering, we obtained a VisEvent-VOS subset suitable for evaluating RGB-E video object segmentation. Since there were previously no models capable of RGB-E video object segmentation,

**Table 1: RGB-T performance on VisT300 test set and VT-UAV test set.**

| | | | | | VisT300 Benchmark | | | | | | |
|---|---|---|---|---|---|---|---|---|---|---|---|
| Method | STM [51] | STCN [9] | STCN-T [73] | HMMN [60] | TBD [12] | CFBI+ [80] | AOT [79] | XMem [8] | XMem-T [73] | VTiNet [73] | X-Prompt (Ours) |
| $\mathcal{J}\&\mathcal{F}$ | 60.4 | 71.4 | 72.3 | 68.3 | 70.5 | 74.1 | 76.8 | 75.7 | 77.9 | 81.9 | **84.2** |
| $\mathcal{J}$ | 57.9 | 74.4 | - | 65.9 | 68.3 | 71.8 | 74.0 | 73.3 | - | 79.2 | **81.7** |
| $\mathcal{F}$ | 62.8 | 73.8 | - | 70.6 | 72.6 | 76.4 | 79.6 | 78.0 | - | 84.5 | **86.7** |
| | | | | | VT-UAV Benchmark | | | | | | |
| Method | RANet [68] | TVOS [87] | SiamMask [64] | D3S [47] | AlphaRefine [72] | STCN [9] | AOT-B [79] | AOT-L-Swin [79] | XMem [8] | VTiNet [73] | X-Prompt (Ours)) |
| $\mathcal{J}\&\mathcal{F}$ | 38.8 | 44.0 | 57.0 | 57.0 | 65.9 | 65.5 | 76.6 | 82.0 | 69.1 | 76.7 | **87.3** |
| $\mathcal{J}$ | 32.2 | 36.9 | 52.9 | 53.4 | 59.9 | 61.0 | 72.1 | 77.5 | 65.1 | 72.9 | **82.8** |
| $\mathcal{F}$ | 45.4 | 51.0 | 61.0 | 60.7 | 71.9 | 69.9 | 81.1 | 86.5 | 73.1 | 80.8 | **91.8** |

**Table 2: RGB-D performance on ARKitTrack-VOS test set.**

| | Only Train with RGB-D Data | | | | | | With RGB Data Pre-Train | | | | |
|---|---|---|---|---|---|---|---|---|---|---|---|
| Method | QDMN [45] | RPCM [71] | AOT [79] | STCN [9] | STCN-D [88] | ARKitVOS [88] | STCN [9] | XMem [8] | AOT-B [79] | AOT-L-Swin [79] | X-Prompt (Ours) |
| $\mathcal{J}\&\mathcal{F}$ | 30.6 | 50.9 | 58.2 | 52.2 | 53.7 | 66.2 | 63.6 | 71.6 | 72.6 | 77.8 | **82.1** |
| $\mathcal{J}$ | 27.6 | 49.2 | 55.5 | 48.9 | 49.8 | 62.5 | 60.2 | 68.5 | 70.0 | 75.0 | **79.4** |
| $\mathcal{F}$ | 33.7 | 52.7 | 62.7 | 56.0 | 57.5 | 69.8 | 66.9 | 74.6 | 75.3 | 80.7 | **84.9** |

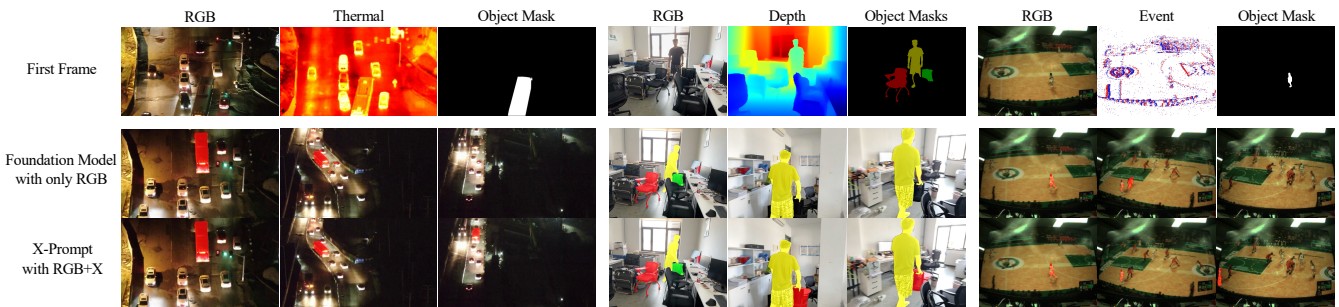

**Figure 4: Qualitative results of RGB-X. X-Prompt effectively utilizes X-modality to address challenging scenarios.**

**Table 3: RGB-E performance on VisEvent-VOS test set.**

| | $\mathcal{J}\&\mathcal{F}$ | $\mathcal{J}$ | $\mathcal{F}$ |
|---|---|---|---|
| STCN [9] | 61.9 | 57.5 | 66.3 |
| AOT-B [79] | 56.3 | 50.5 | 62.0 |
| AOT-L-Swin [79] | 58.6 | 52.2 | 64.9 |
| AOT-L-Swin-E | 59.7 | 54.0 | 65.4 |
| X-Prompt | **67.1** | **61.7** | **72.5** |

we compared our approach with current state-of-the-art VOS models and a fine-tuned AOT-L-Swin model with concatenated RGB and event inputs (row-4). Our method significantly outperformed them, as demonstrated in Table 3.

## 4.3 Ablation Study

**X-Prompt Framework.** To verify the effectiveness of the X-Prompt framework, including foundation model, multi-modal visual prompter, and multi-modal adaptation experts, Tab. 4 presents a series of tests:

the foundation model, using only RGB information, achieves basic object segmentation (row-1); with the addition of the Multi-modal Visual Prompter, we effectively encode X-modality into the prompt, thereby enhancing the intrinsic capabilities of the foundation model (row-2); the best results are achieved when modality experts are involved to adapt the foundation model for specific modality downstream tasks (row-4). We also independently assessed the contribution of the multi-modal adaptation experts (row-3) by simply concatenating RGB and X-modality data, instead of using the proposed prompter.

**Multi-modal Visual Prompter.** We employed different visual prompts for the foundation model to analyze the roles of various modalities and demonstrate the effectiveness of the proposed multi-modal visual prompter. As shown in Tab. 5, RGB contains primary information, and achieving accurate segmentation solely with auxiliary X-modalities is difficult. While thermal modality, in particular, inherently contains relatively more information for object segmentation. The proposed MVP effectively encodes RGB images and

**Table 4: Ablation study of the X-Prompt framework.**

| | VisT300 | | | VT-UAV | | | ARKitTrack | | | VisEvent | | |
|---|---|---|---|---|---|---|---|---|---|---|---|---|
| | $\mathcal{J}\&\mathcal{F}$ | $\mathcal{J}$ | $\mathcal{F}$ | $\mathcal{J}\&\mathcal{F}$ | $\mathcal{J}$ | $\mathcal{F}$ | $\mathcal{J}\&\mathcal{F}$ | $\mathcal{J}$ | $\mathcal{F}$ | $\mathcal{J}\&\mathcal{F}$ | $\mathcal{J}$ | $\mathcal{F}$ |
| Foundation Model | 76.5 | 73.7 | 79.2 | 82.4 | 77.9 | 86.9 | 78.4 | 75.8 | 81.1 | 60.7 | 55.3 | 66.0 |
| with MVP | 80.9 | 78.3 | 83.4 | 83.7 | 79.5 | 87.9 | 79.5 | 76.7 | 82.4 | 62.5 | 57.1 | 67.8 |
| with MAEs | 81.6 | 79.0 | 84.3 | 86.8 | 82.2 | 91.5 | 81.3 | 78.3 | 84.3 | 61.2 | 55.1 | 67.3 |
| with both (X-Prompt) | **84.2** | **81.4** | **87.0** | **87.5** | **83.0** | **91.9** | **82.1** | **79.4** | **84.9** | **67.1** | **61.7** | **72.5** |

**Table 5: Ablation study on the multi-modal visual prompter**

| Prompt | VisT300 | | | VT-UAV | | | ARKitTrack | | | VisEvent | | |
|---|---|---|---|---|---|---|---|---|---|---|---|---|
| | $\mathcal{J}\&\mathcal{F}$ | $\mathcal{J}$ | $\mathcal{F}$ | $\mathcal{J}\&\mathcal{F}$ | $\mathcal{J}$ | $\mathcal{F}$ | $\mathcal{J}\&\mathcal{F}$ | $\mathcal{J}$ | $\mathcal{F}$ | $\mathcal{J}\&\mathcal{F}$ | $\mathcal{J}$ | $\mathcal{F}$ |
| only RGB | 76.5 | 73.7 | 79.2 | 82.4 | 77.9 | 86.9 | 78.4 | 75.8 | 81.1 | 60.7 | 55.3 | 66.0 |
| only X | 61.6 | 60.7 | 62.4 | 38.9 | 33.2 | 44.6 | 32.7 | 31.7 | 33.8 | 22.9 | 19.4 | 26.3 |
| Cat(RGB, X) | 81.6 | 79.0 | 84.3 | 86.8 | 82.2 | 91.5 | 81.3 | 78.3 | 84.3 | 61.2 | 55.1 | 67.3 |
| MVP | **84.2** | **81.4** | **87.0** | **87.5** | **83.0** | **91.9** | **82.1** | **79.4** | **84.9** | **67.1** | **61.7** | **72.5** |

**Table 6: Ablation study on the multi-modal adaptation experts.**

| Adaptation | Learnable Parameters | VisT300 | | | VT-UAV | | | ARKitTrack | | | VisEvent | | |
|---|---|---|---|---|---|---|---|---|---|---|---|---|---|
| | | $\mathcal{J}\&\mathcal{F}$ | $\mathcal{J}$ | $\mathcal{F}$ | $\mathcal{J}\&\mathcal{F}$ | $\mathcal{J}$ | $\mathcal{F}$ | $\mathcal{J}\&\mathcal{F}$ | $\mathcal{J}$ | $\mathcal{F}$ | $\mathcal{J}\&\mathcal{F}$ | $\mathcal{J}$ | $\mathcal{F}$ |
| Frozen | 0M, 0% | 80.9 | 78.3 | 83.4 | 83.7 | 79.5 | 87.9 | 79.5 | 76.7 | 82.4 | 62.5 | 57.1 | 67.8 |
| Full FT | 106.5M, 100% | 76.1 | 73.6 | 78.7 | 72.2 | 67.4 | 77.0 | 62.2 | 59.3 | 65.1 | 40.0 | 35.9 | 44.0 |
| Adapter | 0.2M, 0.2% | 83.5 | 80.9 | 86.1 | 86.2 | 82.0 | 90.5 | 80.1 | 77.3 | 83.0 | 64.3 | 58.8 | 69.9 |
| LoRA | 0.8M, 0.7% | 83.0 | 80.4 | 85.6 | 85.6 | 81.3 | 89.9 | 80.4 | 77.7 | 83.2 | 62.7 | 57.4 | 68.0 |
| MAEs | 2.3M, 2.1% | **84.2** | **81.4** | **87.0** | **87.5** | **83.0** | **91.9** | **82.1** | **79.4** | **84.9** | **67.1** | **61.7** | **72.5** |

**Table 7: Ablation study on the number of modality experts.**

| Experts Number | Learnable Parameters | VisT300 | | | VT-UAV | | | ARKitTrack | | | VisEvent | | |
|---|---|---|---|---|---|---|---|---|---|---|---|---|---|
| | | $\mathcal{J}\&\mathcal{F}$ | $\mathcal{J}$ | $\mathcal{F}$ | $\mathcal{J}\&\mathcal{F}$ | $\mathcal{J}$ | $\mathcal{F}$ | $\mathcal{J}\&\mathcal{F}$ | $\mathcal{J}$ | $\mathcal{F}$ | $\mathcal{J}\&\mathcal{F}$ | $\mathcal{J}$ | $\mathcal{F}$ |
| 0 | 0M, 0% | 80.9 | 78.3 | 83.4 | 83.7 | 79.5 | 87.9 | 79.5 | 76.7 | 82.4 | 62.5 | 57.1 | 67.8 |
| 1 | 1.5M, 1.4% | 83.7 | 81.0 | 86.5 | 86.8 | 82.3 | 91.3 | 80.7 | 77.8 | 83.5 | 65.4 | 59.7 | 71.2 |
| 2 | 2.3M, 2.1% | **84.2** | **81.4** | **87.0** | **87.5** | **83.0** | **91.9** | 81.3 | 78.5 | 84.1 | **67.1** | **61.7** | **72.5** |
| 3 | 3.2M, 2.9% | 81.6 | 79.0 | 84.3 | 86.8 | 82.2 | 91.5 | **82.1** | **79.4** | **84.9** | 65.0 | 59.7 | 70.4 |
| 4 | 4.1M, 3.7% | 82.6 | 80.1 | 85.1 | 86.3 | 82.1 | 90.6 | 80.1 | 77.4 | 82.9 | 64.6 | 59.2 | 70.1 |
| 5 | 4.9M, 4.4% | 83.5 | 80.8 | 86.1 | 86.3 | 82.0 | 90.6 | 80.6 | 77.7 | 83.5 | 64.2 | 58.5 | 69.8 |

X-modality complementarily, forming a visual prompt that enables the foundation model to achieve accurate segmentation.

**Multi-modal Adaptation Experts.** First, we implemented various approaches for adapting the foundation model to downstream X-modality tasks, as illustrated in Tab. 6. "Frozen" refers to training only the multi-modal visual prompter (MVP) while keeping the entire foundation model frozen (row-1). Full parameter fine-tuning leads to model degradation and catastrophic forgetting issues (row-2). Adapter and LoRA techniques allow the foundation model to adapt without degradation (row-3), demonstrating the feasibility and effectiveness of the foundation model - prompt - adaptation paradigm (rows 3 and 4). The proposed Multi-modal Adaptation Experts achieve the best performance by introducing additional independent experts during training to learn multi-modal knowledge. Next, as shown in Tab. 7, we experimented with the number of experts employed and found that introducing two experts achieves the best modality adaptation for most RGB-X tasks, except for the RGB-D tasks in the ARKitTrack scenario, where three experts yield better performance.

## 5 CONCLUSION

In this work, we introduce X-Prompt, the first universal framework for multi-modal RGB+X video object segmentation tasks, to address the issue of model generalization degradation due to limited multi-modal data and reduces redundant research effort and computational deployment costs associated with task-specific designs. Following the pre-training of an RGB VOS foundation model with robust segmentation capabilities and generalization, X-Prompt utilizes the X-modality to prompt and adapt the foundation model for various downstream multi-modal tasks, employing our proposed Multi-modal Visual Prompter and Multi-modal Adaptation Experts. Extensive experiments on 3 tasks across 4 benchmarks demonstrate the effectiveness and generalization of X-Prompt.

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
