# OpenReview forum: "X-Prompt: Multi-modal Visual Prompt for Video Object Segmentation"
_acmmm.org/ACMMM/2024/Conference — MM2024 Poster_

### Official Review · Reviewer_GLJH · 2024-05-18

**Rating:** 4
**Confidence:** 3

**Summary:**

The paper proposes a unified framework (X-Prompt) for semi-supervised multi-modal video object segmentation tasks, including RGB-Thermal, RGB-Depth and RGB-Event. Meanwhile, it proposes the Multi-modal Visual Prompter (MVP) for early multi-modal fusion and introduces the Multi-modal Adaptation Expert (MAEs) for parameter-efficient transfer learning. It conducts extensive experiments on 3 tasks across 4 benchmarks and achieves impressive performance.

**Strengths:**

- Novel paradigm: pre-training for single modal (RGB) and parameter-efficient transfer learning for multi-modal (RGB+X).
- Extensive experiments on 3 tasks across 4 benchmarks and impressive performance on all the benchmarks.
- Comprehensive and valuable ablation experiments.

**Limitations:**

**METHOD**
- The novelty of MVP is limited, just following general multi-modal fusion strategies. And why only enhance RGB feature by Spatial-Channel
Attention Weights instead of both RGB feature and X-modal feature?
- Confusing symbols. (1) What are the $z_t^0$ and $z_r^0$ in Eq.(2) meaning?  (2) $m_r$ appears at Eq.(3), but is defined at Line-390. I don't know the differences between $m_r$ and $M_r$ until I read the definition in Line-390. (3) Does the $z_t^L = F(z_t^0, z_r^0, M_r)$ mean the feature of target frame from L-th layer? But it seems that it outputs features of both reference frame and target frame for each layer ($z^l = F^{(l)}(z^{l-1}, m_r)$).
- The detailed architecture of Downsample Patch Embedding is several cascaded convolution layers? Can Mask Decoder really benefit from features extracted by shallow convolutional layers?


**Ablation Study:**

- In Table 4, row-2 has 0M learnable parameters, since the results is same as row-1 in Table.6. Thus, I guess the row-1 in Table 4 also has 0M learnable parameters. Meanwhile, the row-1 and row-2 in Table 5 don’t have learnable parameters.

Table 5 row-1: Frozen Foundation Model + RGB

Table 5 row-2: Frozen Foundation Model + X
Table 5 row-3: Frozen Foundation Model + MAEs + Concat (RGB & X) (same as row-3 in Table 4)

Table 5 row-4: Frozen Foundation Model + MAEs + MVP(RGB & X)

In Table 5, row-1 and row-2 cannot be fairly compared to the results from row-3 and row-4. I would like to know the results of:

Frozen Foundation Model + MAEs + RGB

Frozen Foundation Model + MAEs + X

**Minor Errors:**

- Page 5, Line 518: extra “,”  and wrong symbol $z_{rgb}^{16 \times 16}$;

- For Figure 2, I don't understand how to use the reference mask;

- For Figure 3, the lines extending down from the RGB and X encoders are hard to understand.

**Suitability:**

3

---

### Official Review · Reviewer_r5ea · 2024-05-23

**Rating:** 5
**Confidence:** 3

**Summary:**

This paper presents X-Prompt, a new multi-modal video object segmentation (VOS) framework for RGB-Thermal, RGB-Depth, and RGB-Event scenarios. First, X-Prompt is used for traditional VOS tasks by training a foundation model. Immediately after that, the proposed method constructs the Multi-modal Visual Prompter and introduces the Multi-modal Adaptation Experts for adapting the information of each modality. The manuscript shows extensive experiments and ablations. X-Prompt is superior to existing VOS methods.

**Strengths:**

1. The manuscript is well organized.

2. The motivation for multi-modal VOS is clear and well-illustrated in the manuscript.

3. This work is solid, and the experimental results are promising.

**Limitations:**

1. This paper uses ConvMAE as the backbone but does not specify what version. Small, Base, or Large? Also, ConvMAE uses pre-training of MAE and then fine-tuning on IN1K. How well would the proposed method work if it used traditional ViT?

2. Section 2.1, titled Video Object Segmentation, only surveys the SVOS methods. The authors should discuss and describe the unsupervised VOS methods or change the title to Semi-supervised VOS.

3. The best performance for the ARKitTrack dataset in Table 7 is an Expert Number of 3, which gives the learnable parameters of 3.2M instead of 2.3M in Table 6.

4. The foundation model's video sequence in Figure 3 of the supplementary material is incorrect.

5. Typos.
1) In L26, ‘Multi-modal Adaptation Expert’ should be ‘Multi-modal Adaptation Experts’.
2) In L228, ‘And’ should be ‘and’.
3) In L635, ‘The Foundation model’ should be ‘The foundation model’.

**Suitability:**

3

---

### Official Review · Reviewer_vNcn · 2024-05-23

**Rating:** 5
**Confidence:** 3

**Summary:**

This paper introduces X-Prompt, a universal framework for multi-modal video object segmentation (VOS) that supports RGB-Thermal, RGB-Depth, and RGB-Event modalities. X-Prompt pre-trains a foundation model on RGB data and adapts it to multi-modal tasks using the Multi-modal Visual Prompter (MVP) and Multi-modal Adaptation Experts (MAEs) to effectively handle limited data scenarios. Extensive experiments across multiple benchmarks demonstrate that X-Prompt consistently outperforms traditional fine-tuning methods, achieving state-of-the-art performance in multi-modal VOS tasks.

**Strengths:**

1. The proposed X prompt is novel and has good practical usage. Arbitrary modality fusion in video object segmentation is also very important and less explored in the past. I think the motivation is significant to the community,

2. MVP and MAEs are well designed to tackle the integration and adaptation of the multi-modal data effectively.

3. The proposed approach is verified to be effective among different datasets. Comprehensive ablation studies are done to illustrate the efficay of each designed component.

**Limitations:**

1. Lack of comparison with arbitrary modality fusion approaches, e.g., [1].
[1] Zhang, J., Liu, R., Shi, H., Yang, K., Reiß, S., Peng, K., ... & Stiefelhagen, R. (2023). Delivering arbitrary-modal semantic segmentation. In Proceedings of the IEEE/CVF Conference on Computer Vision and Pattern Recognition (pp. 1136-1147).

2. The RGB-T performance are verified on two datasets while the RGB-D and RGB-E performances are only verified on one dataset. It lacks of justification of the generalizability on  RGB-D and RGB-E tasks.

3. Can the model be adapted into image based object segmentation task? Will it deliver comparable performances?

4. Lack of the analyses of the limitations and the failure cases. The authors are encouraged to enrich these analyses in their main paper.

**Suitability:**

3

---

### Official Review · Reviewer_8r6x · 2024-06-01

**Rating:** 3
**Confidence:** 2

**Summary:**

The paper introduces a universal framework named X-Prompt designed for multi-modal video object segmentation tasks, such as RGB-Thermal, RGB-Depth, and RGB-Event. This framework aims to address challenges faced by traditional video object segmentation (VOS) methods in scenarios with extreme illumination, rapid motion, and background distractions.

**Strengths:**

1.The paper introduces a novel framework, X-Prompt, which is a universal solution for multi-modal video object segmentation tasks. This approach is designed to handle various challenging scenarios like extreme illumination, rapid motion, and background distraction.
2. X-Prompt leverages a foundation model pre-trained on RGB data and adapts it to multi-modal tasks using additional modalities with limited data, which can be more efficient than training from scratch and helps maintain the model's generalization ability.
3. X-Prompt achieves state-of-the-art performance on multiple benchmarks for RGB-Thermal, RGB-Depth, and RGB-Event tasks, demonstrating its effectiveness.

**Limitations:**

1. X-Prompt might be more complex compared to traditional VOS methods, which could potentially make it harder to implement and maintain. And there's an inherent risk of overfitting when fine-tuning with limited data, especially in complex real-world scenarios. And the ability of X-Prompt to generalize to completely new and unseen scenarios still require further validation.
2. Give more details about the computational cost, X-Prompt introduces additional modalities, which might lead to higher computational costs compared to simpler models.
3. A thorough check for grammatical,spelling errors and figures is recommended to polish the paper.

**Suitability:**

2

---

### Meta-Review · Area_Chair_hK7p · 2024-07-01

**Recommendation:** Accept (Poster)
**Confidence:** 5

**Metareview:**

The authors did a good rebuttal. The reviewers unanimously recommend acceptance in the final rating. After checking the rebuttal, the review, and the paper, the AC agrees with this assessment.